# Sound Based Fault Diagnosis Method Based on Variational Mode Decomposition and Support Vector Machine

**Xiaojing Yin** [1], **Qiangqiang He** [1], **Hao Zhang** [2], **Ziran Qin** [2] and **Bangcheng Zhang** [1,3,*]

[1] Mechanical and Electrical Engineering, Changchun University of Technology, Changchun 130012, China; yinxiaojing2011@163.com (X.Y.); john_ho2021@163.com (Q.H.)
[2] Changchun Faway Automobile Mirror System Co., Ltd., Changchun 130011, China; zhanghao@fawayqcj.com (H.Z.); qinziran@fawayqcj.com (Z.Q.)
[3] Mechanical & Automotive Engineering, Changchun Institute of Technology, Changchun 130103, China
[*] Correspondence: zhangbangcheng@ccut.edu.cn

**Abstract:** In industry, it is difficult to obtain data for monitoring equipment operation, as mechanical and electrical components tend to be complicated in nature. Considering the contactless and convenient acquisition of sound signals, a method based on variational mode decomposition and support vector machine via sound signals is proposed to accurately perform fault diagnoses. Firstly, variational mode decomposition is conducted to obtain intrinsic mode functions. The fisher criterion and canonical discriminant function are applied to overcome the fault diagnosis accuracy decline caused by intrinsic mode functions with multiple features. Then, the fault features obtained from these intrinsic mode functions are chosen as the final fault features. Experiments on a car folding rearview mirror based on sound signals were used to verify the superiority and feasibility of the proposed method. To further verify the superiority of the proposed model, these final fault features were taken as the input to the following classifiers to identify fault categories: support vector machine, k-nearest neighbors, and decision tree. The model support vector machine achieved an accuracy of 95.8%, i.e., better than the 95% and 94.2% of the other two models.

**Keywords:** sound signals fault diagnosis; variational mode decomposition; intrinsic mode functions selection; multiple feature extraction; support vector machine

## 1. Introduction

Benefiting from the easy availability of sound signal acquisition in industry, fault diagnosis methods based on sound signals have played an important role in recent years [1]. Generally, sound signals from industry have the characteristics of being non-smooth and non-linear, so it is a challenging task to identify types of failure of mechanical components [2]. Various signal processing methods are effective at overcoming this problem [3]. For example, a fault diagnosis method combining signal processing and intelligent classification models has been explored [4,5]. However, traditional fault diagnosis methods based on sound signals suffer from accuracy declines caused by multiple features. Therefore, it is essential to put forward an effective and feasible fault diagnosis method based on sound signals.

In the past decades, fault diagnosis methods have mainly focused on threshold methods, expert systems based on subjective experience, neural networks, and signal processing methods. In the literature [6], a threshold fault diagnosis method based on dynamic time warping (DTW) was proposed and achieved high recognition accuracy. However, this method can only identify whether the device is faulty. A fault diagnosis method based on expert knowledge was proposed for mechanical components [7], and it acquired good interpretability. In the face of practical engineering problems, how to better combine data with expert knowledge may be a difficult problem. Neural networks can extract deep fault

features due to their deep structure construction [8]. Because neural networks usually need multiple training sets, such models will be limited in some practical aspects.

A fault diagnosis method combining signal processing and intelligent classification provides the possibility to address the problems [9]. A Fuzzy c-means and similarity function-based fault diagnosis method was proposed; it was shown to be able to precisely identify the fault types of turnout [10]. A signal processing method based on spectrum analysis and support vector machine (SVM) was proposed and achieved excellent fault diagnosis effectiveness [5]. Otherwise, the empirical mode decomposition (EMD) signal processing method acquires excellent results in the case of non-smooth and non-linear signals and is widely used. However, there are several problems with sifting stop criterion, end effects, and mode mixing in the decomposition signals [11]. In order to overcome the problems of EMD, the variational mode decomposition (VMD) signal decomposition method was proposed [12].

Due to the effective of dimensionless features in the characterization of fault information, such approaches have been widely applied for fault diagnoses [13]. In order to extract the useful features, selecting the optimal intrinsic mode functions (IMFs) is a feasible method. The authors of [14] proposed the selection of the optimal IMFs based on the kurtosis-max principle, and then extracted dimensionless metrics from these IMFs. Considering the fact that the single kurtosis principle may ignore higher energy IMFs, the optimal IMFs selection method based on the dual principle of kurtosis and energy spectrum was applied and achieved excellent fault diagnosis results [15]. The IMFs were selected as the input of the convolutional neural network by HHT processing and quantization, and were used for fault diagnosis [16]. The above optimal IMFs selection methods are only for a single fault signal, and the comprehensive selection of optimal IMFs is not explored. Otherwise, a decrease in the diagnosis accuracy will occur when dimensionless features with high correlation and multiple features are used for fault diagnoses [17,18].

To address this problem, the authors of [19] used principal component analysis to reduce the dimensionality of features. The authors of [20] used sensitivity to characterize the importance of the features, and then obtained a ranking of the features with high sensitivity as the final fault features. A method based on the fisher criterion (FC) and self-weight criterion was conducted to search for the useful features [21]. The authors of that paper realized the reduction and extraction of metrics from the perspective of selecting the optimal IMFs and solved the problem whereby the high-dimensional features in complex samples leads to decreased fault diagnosis accuracy. Otherwise, based on the advantages of VMD in processing non-smooth nonlinear, the superiority of SVM in small samples and the contactless and convenient acquisition of sound signals, in our work, a fault diagnosis method via sound based on VMD-SVM and FC for IMFs selection is proposed.

The main steps of the proposed fault diagnosis method are as follows. Firstly, the superiority of the VMD signal decomposition method was confirmed compared with EMD in the processing of nonlinear, non-stationary signals. Subsequently, the IMFs were obtained through the VMD signal processing approach from sound signals. Then, the optimal IMFs were selected by FC, and a discriminant function based on the importance of the feature was acquired to determine the top-ranked features of each fault type. These features were used as the final fault features as the input for the classification model. Based on our experimental data, the superiority of the proposed method for optimal IMF selection based on VMD-SVM is demonstrated.

The rest of this study is structured as follows. Section 2 introduces the method proposed in this work. Section 3. describes the collection process and the experimental results. Finally, The conclusions of this paper are summarized in Section 4.

## 2. Materials and Methods

*2.1. Diagnosis Approach Based on Sound Signals*

Based on the proposed IMF selection method with VMD and SVM, a fault diagnosis method for mechanical components is proposed. A diagram describing the method is given as Figure 1.

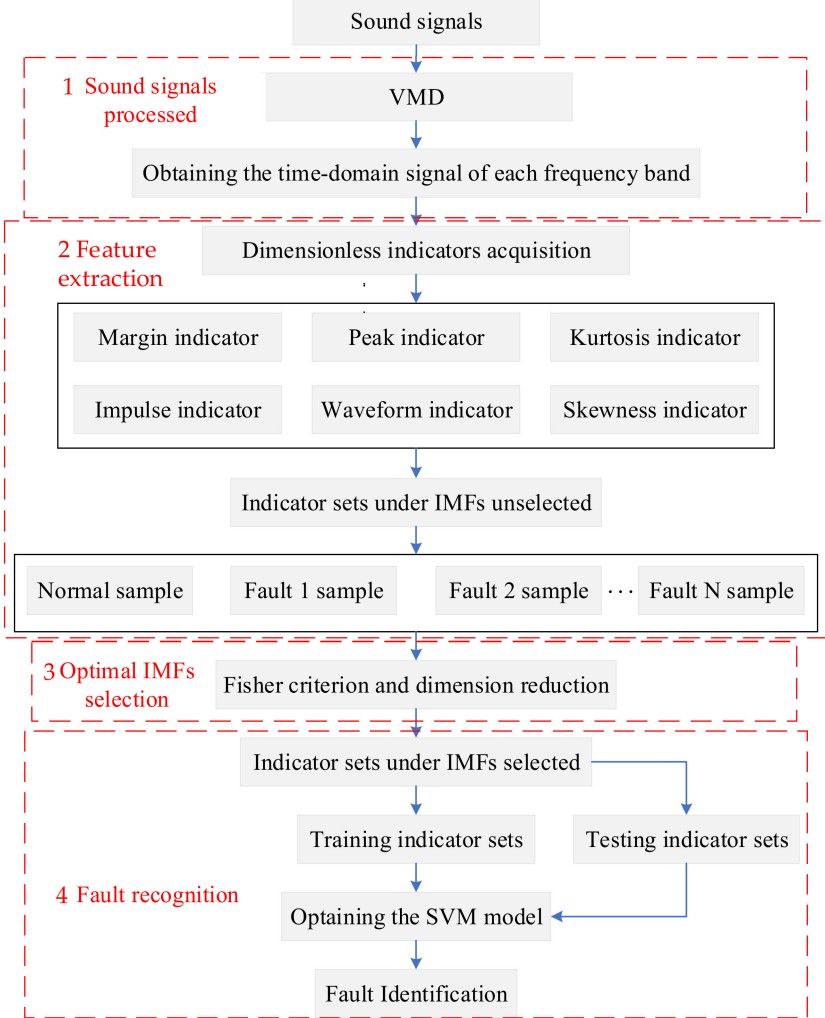

**Figure 1.** Diagram of the proposed method.

The main steps are as follows:

1. Sound signals processed

The sound signals of mechanical components are processing through VMD to acquire a set of IMFs.

2. Feature extraction

Six dimensionless features are extracted from the unselected IMFs.

3. Optimal IMF selection

The top-ranked IMFs are obtained by fisher FD and the optimal IMFs that can fully characterize fault information are selected through the canonical discriminant function (CDF).

4. Fault recognition

In this step, SVM is used to verify the effectiveness of the proposed fault diagnosis method based on the sound signal.

### 2.2. Variational Mode Decomposition

VMD is a non-recursive, adaptive signal processing technique. The original signal is split into IMFs in the VMD architecture, and each obtained IMF is compressed around the matching center frequency. The bandwidth of the component can be evaluated using constrained variation. The constrained variation is expressed as [22]:

$$
\min_{\{u_k\}\{\omega_k\}} \left\{ \sum_k \left\| \partial_t \left[ \left( \delta(t) + \frac{j}{\pi t} \right) * u_k(t) \right] e^{-j\omega_k t} \right\|_2^2 \right\}
$$
$$
s.t. \sum_k u_k = f
$$
(1)

where $u_k$ is the $k$ th component of the signal and $\omega_k$ denotes the center frequency of the signal whose kth component, $f$, is the original signal; $\delta$ is the Dirac distribution, $t$ is the time script, and $*$ denotes convolution.

By introducing a quadratic penalty and Lagrangian multipliers, the above constrained optimization problem can be expressed as follow [22]:

$$
L\left(\{u_k\}, \{\omega_k\}, \lambda\right) = \alpha \sum_k \left\| \partial_t \left[ \left( \delta(t) + \frac{j}{\pi t} \right) * u_k(t) \right] e^{-j\omega_k t} \right\|^2
$$
$$
+ \left\| f(t) - \sum_k u_k(t) \right\|_2^2 + \left\langle \lambda(t), f(t) - \sum u_k(t) \right\rangle
$$
(2)

where $\alpha$ denotes the balancing parameter of the data-fidelity constraint.

Parameter $\alpha$ is inversely proportional to the bandwidth of the IMFs. If $\alpha$ is too small, the mode bandwidth is wide; however, decreasing the bandwidth by increasing $\alpha$ comes with the risk of not properly capturing the correct center frequency. There are too few IMFs for the number of IMFs, $k$, resulting in inadequate data segmentation. Some components are reused in subsequent patterns, while others are eliminated as "noise". In this study, we established the parameters of IMF and $k$, keeping this in mind.

Equation (2) is then solved with the alternate direction method of multipliers, and then all the modes gained from solutions in the spectral domain are written as [22]:

$$
u_k^{n+1}(\omega) = \frac{f(\omega) - \sum\limits_{i \neq k} u_i(\omega) + \frac{\lambda(\omega)}{2}}{1 + 2\alpha(\omega - \omega_k)^2}
$$
(3)

where, $f(\omega)$, $u_i(\omega)$, $\lambda(\omega)$ and $u_k^{n+1}(\omega)$ are the Fourier transforms of $y(t)$, $y_i(t)$, $\lambda(t)$, $y_i^{n+1}(t)$ and $\omega$ denotes the center frequency of the signal. It is noted that Equation (3) contains the Wiener filter structure. The component in the time domain can be obtained from the real part of the inverse Fourier transform of the filtered signal.

### 2.3. Dimensionless Indicators

Mechanical failures are common [23], and may be seen in sound signals as changes in time-domain amplitude and frequency. These features can be rapidly obtained. In this paper, six time-domain feature parameters are extracted [24]. As shown in Table 1.

**Table 1.** Time-domain features.

| Feature | Equation |
|---|---|
| Mean | $x = \frac{\sum_{i=1}^{N}|x(i)|}{N}$ |
| Skewness | $x_{ske} = \frac{\sum_{i=1}^{N}(x(i)-\bar{x})^3}{(N-1)x_{std}^3}$ |
| Kurtosis | $x_{kur} = \frac{\sum_{i=1}^{N}(x(i)-\bar{x})^4}{(N-1)x_{std}^4}$ |
| Peak | $x_p = \max|x(i)|$ |
| Impulse factor | $x_{if} = \frac{x_p}{(1/N)\sum_{i=1}^{N}|x(i)|}$ |
| Margin factor | $x_{mf} = \frac{x_p}{\left((1/N)\sum_{i=1}^{N}\sqrt{|x(i)|}\right)^2}$ |

Note: $N$ is the length of the corresponding sound signal, $x$ is the time-domain signal variable.

*2.4. Fisher Discriminant*

Fisher's discriminant criterion takes into account the interclass information in the training samples. Its main purpose is to select features with larger and smaller intraclass distances as the distinguishing features, and to reflect the distinguishing degree of the features by the value of the discriminant function. The larger the value of the discriminant function, the more effective the corresponding feature for classification [25].

The Fisher criterion is simple and efficient and is used in this section to select the best set of features. Thus, there are 120 sets of feature matrices in three categories, each of which corresponds to three four-valued vectors, for the Fisher's discriminant analysis. The calculation process is as follows [25].

Step 1: Calculate the average of the feature vectors in each category

$$\mu_i = \sum_{en \in C_i} en/n_i, (i = 1, 2, \cdots, K) \tag{4}$$

where $C_i$ $(i = 1, 2, \cdots, K)$ denotes the different working conditions, denotes the feature vector, and $n_i$ denotes the number of samples corresponding to each working condition.

Step 2: Calculate the mean of each sample

$$\mu = \sum_{i=1}^{K}(n_i\mu_i) / \sum_{i=1}^{K} n_i \tag{5}$$

Step 3: Calculating intra-class dispersion

$$S_w = \sum_{i=1}^{K}\sum_{en \in C_i}\|en-\mu_i\|^2 \tag{6}$$

where $\| en - \mu_i \| = \sqrt{(en - \mu_i)^T(en - \mu_i)}$ denotes the distance between vectors $en$ and $\mu_i$.

Step 4: Calculate the dispersion between classes

$$S_b = \sum_{i=1}^{K} n_i\|\mu_i - \mu^2\| \tag{7}$$

where $\| \mu_i - \mu \| = \sqrt{(\mu_i - \mu)^T(\mu_i - \mu)}$ denotes the distance between vectors $\mu$ and $\mu_i$.

Step 5: Calculate the value of the discriminant function $J$

$$J = \frac{S_b}{S_w} \tag{8}$$

where $J$ is a criterion for selecting the optimal IMFs. It can characterize the sensitivity of the sample used for classification. A higher value of $J$ indicates better discrimination between the samples.

*2.5. SVM*

In Figure 2, circles and squares each represent one sample from the sample set.

$$T = \{(x_1, y_1), \cdots, (x_i, y_i)\} \tag{9}$$

where $x_i \in X = R^n$, $y_i \in Y = \{1, -1\}$, $i = 1, 2, \cdots, n$.

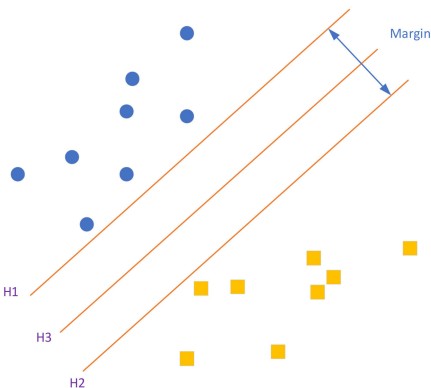

**Figure 2.** The principle of SVM.

The role of SVM is to build an optimal hyperplane that can separate these two types of samples and maximize the interval. In the figure, H is the classification line where the samples are completely separated, and H1 and H2 pass through the nearest distance from H in their respective sample points, remaining parallel to H. The interval between H1 and H2 serves to satisfy the maximum condition [26].

According to the analytic principle [27], the expression of classification line H can be set as

$$mx + b = 0, \ y \in \{1, -1\} \tag{10}$$

$$y_i[mx_i + b] \geq 0, \ i = 1, 2, \cdots, na \tag{11}$$

This leads to a classification interval:

$$H = \frac{2}{\|m\|} \tag{12}$$

It is obvious that $\max(H)$ is equivalent to $\min(\| m \|)$. As such, the classification line minimizes $\| m \|^2/2$. The minimum classification line is optimal. When this classification case is realized on a higher dimensional space, the SVM can control the classification interval to control the generalization ability of the algorithm.

## 3. Rearview Mirror Folding Sound Signal Experiment

*3.1. Results and Analysis*

Sound signals were obtained using an audio recording device. An Audi A6L was used. Two of the experiments were conducted in a semi-anechoic chamber. The signal sampling interval was 20 ms, the collection device model was SQobold, and the model of the semi-anechoic chamber was ZY-2XSS200.

The experimental setup is shown in Figure 3. The first two experiments were carried out in a semi-muffled laboratory, so the signals obtained had little influence from the environment, whereas the third experiment was carried out in the main laboratory. Table 2 shows the operating conditions of the three mirrors and the number of samples, while

Figure 4 shows the time domain diagrams from three rearview mirror folding experiments. The folding function of the rearview mirror was monitored without contact.

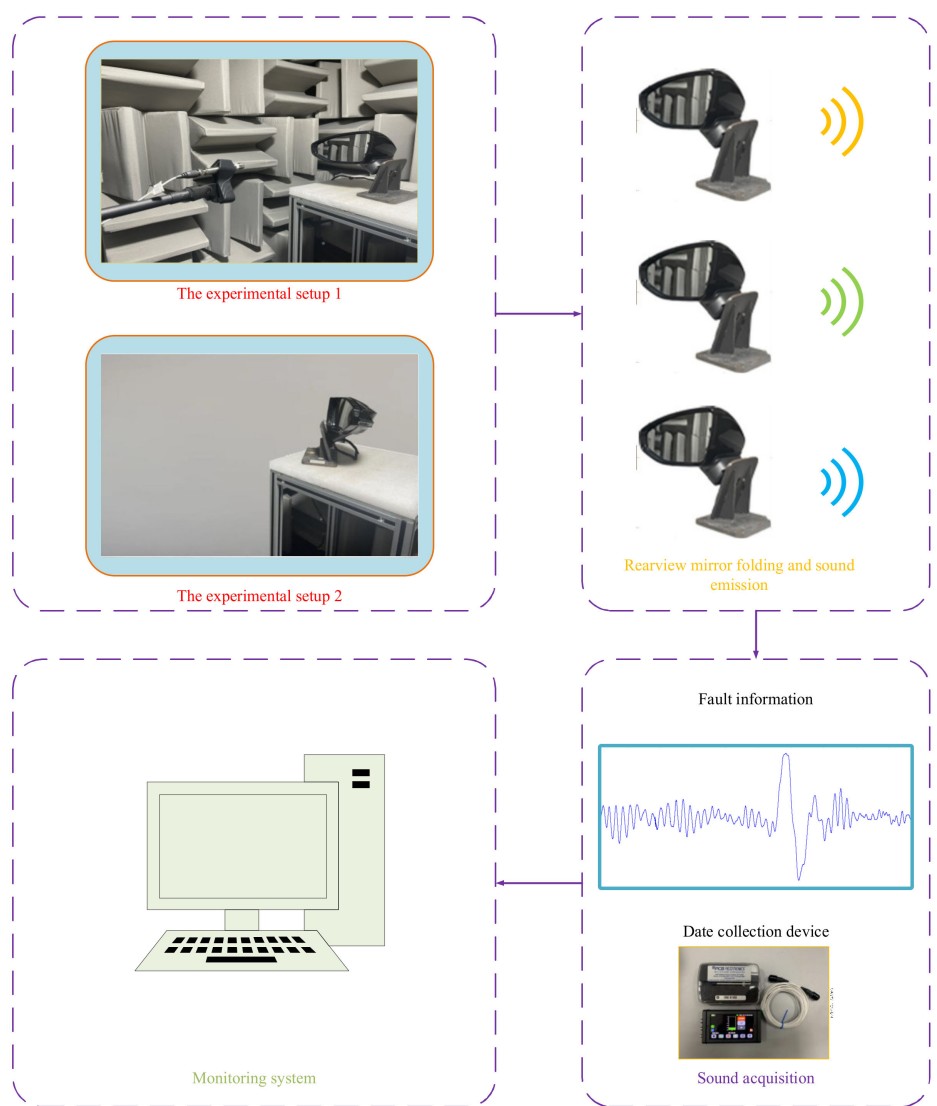

**Figure 3.** Experimental setup. Semi-muffled laboratory (experimental setup 1), and laboratory (experimental setup 2).

**Table 2.** Description of the Working conditions.

| The Serial Number | Descriptions | Sample |
|:---:|:---:|:---:|
| 1 | Abnormal sound of rearview mirror folding in the semi-muffled laboratory | 40 |
| 2 | Normal sound of rearview mirror folding in a semi-muffled room | 40 |
| 3 | Normal sound of rearview mirror folding in the laboratory | 40 |
| | Total | 120 |

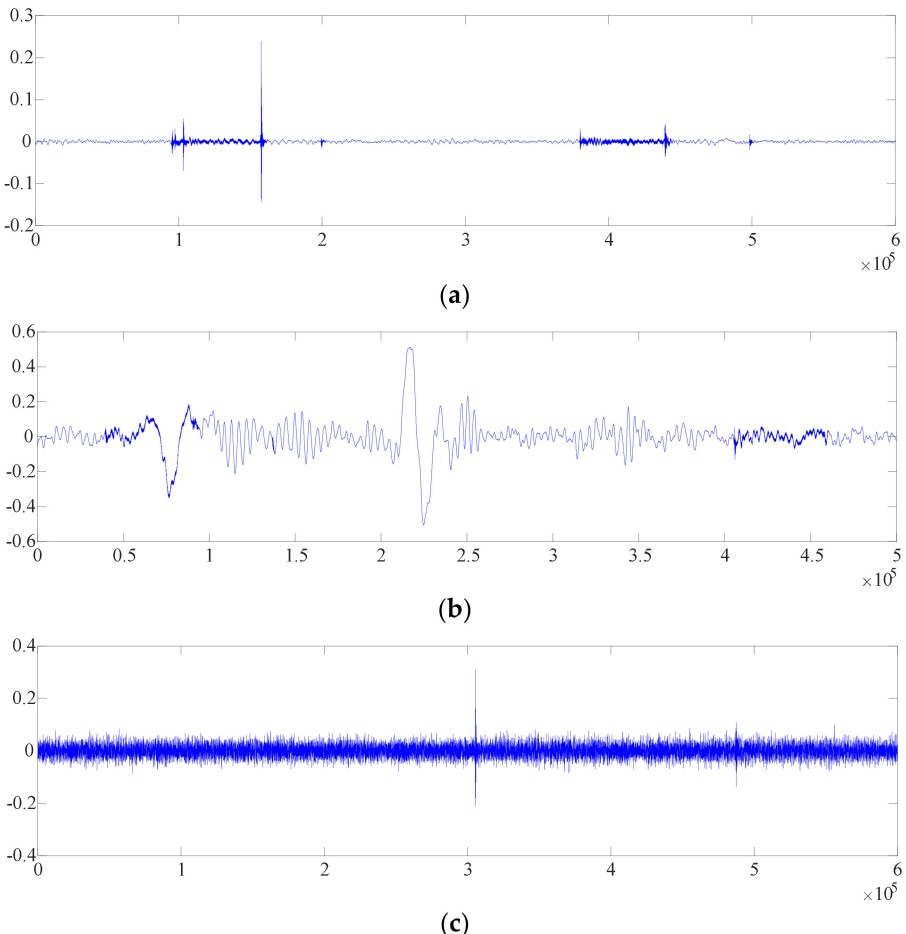

**Figure 4.** Time domain signals for the three operating conditions. Number of samples on the *x*-axis, magnitude on the *y*-axis, (**a**): Abnormal sound of rearview mirror, (**b**): Normal sound of rearview mirror, (**c**): Normal sound of rearview mirror under routine laboratory conditions.

The decomposition, using EMD and VMD, of anechoic chamber noise signals was utilized to extract the IMF components, as illustrated in Figure 5a,b. In terms of the number of dissected IMF components, EMD produces nine, whereas VMD's four-level decomposition yielded four. VMD decomposition decreases the computing size as well as the computation time. Notably, the IMF components dissected by EMD appear to be mode aliased in Figure 5a. VMD divides the amplitude of high frequency components into smaller segments, which aids in classification error reduction [22]. As a result, VMD outperforms EMD in signal processing.

The distribution of the features in the three different signal samples is shown in Scheme 1a–c. The features extracted from the IMFs under each sample have different diagnostic capabilities for the three different signal types. Ideally, the extracted features should have good classification discrimination ability when the features of the three types of signals are clearly separated in the plane space. In contrast, a weak discrimination ability is indicated when there is more overlap in the features of each type. Scheme 1d has more overlap in the distribution of each type of feature, which will be detrimental to the fault identification. As shown in Figure 6, the recognition accuracy of SVM was only 61%.

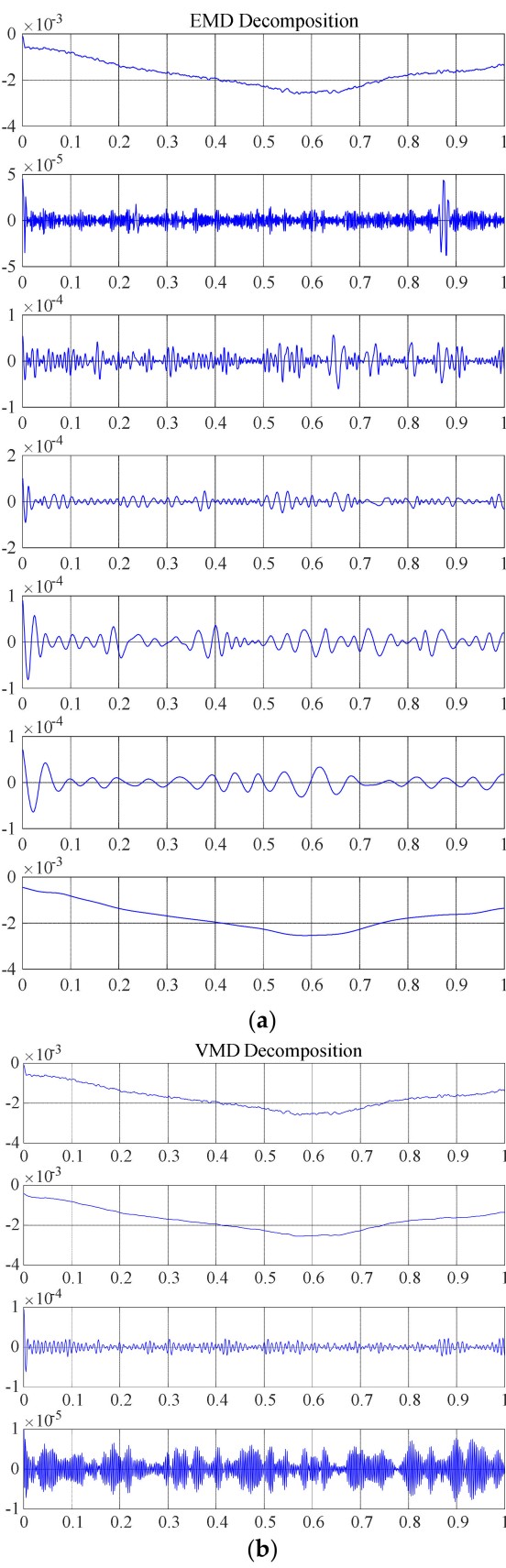

**Figure 5.** The decomposed signal and the corresponding spectrum of the sound signal (Sample 1) processed by EMD (**a**) and VMD (**b**), respectively. Number of samples on the *x*-axis, magnitude on the *y*-axis.

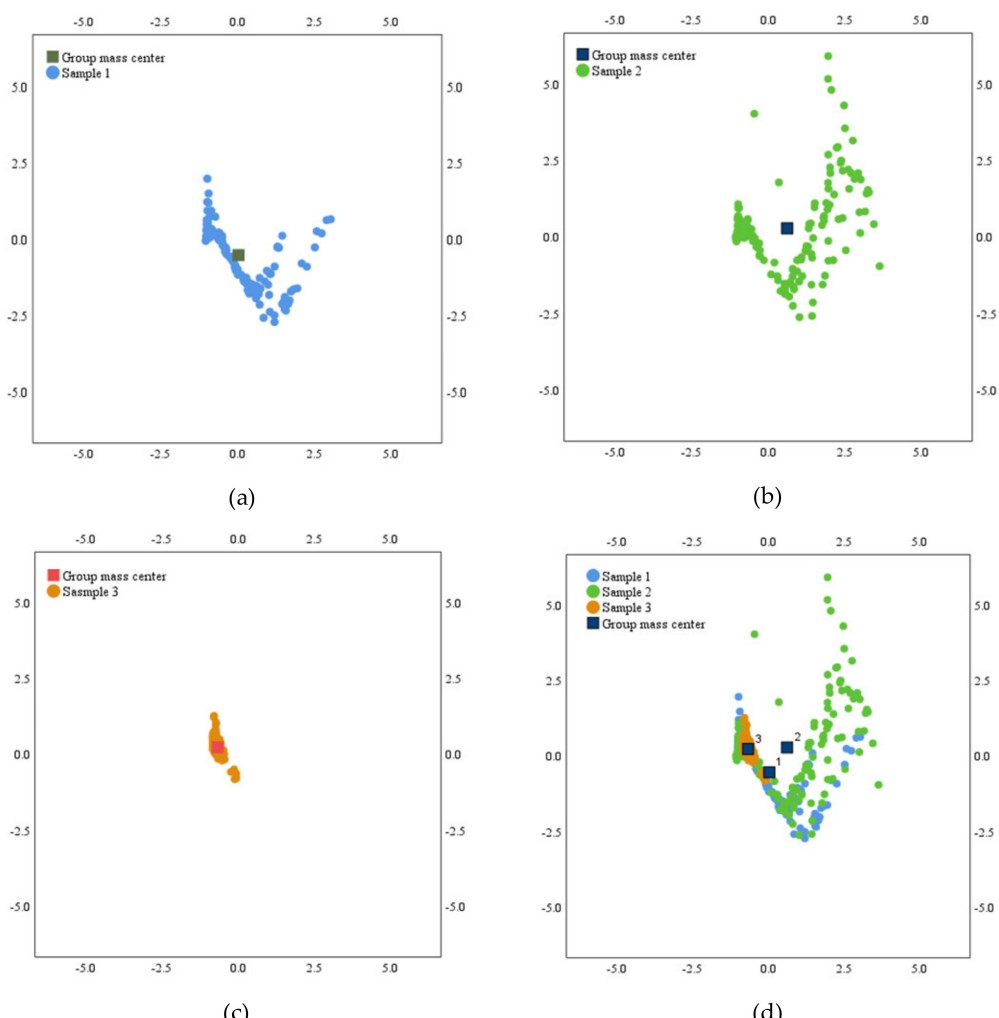

**Scheme 1.** The distribution of unselected IMFs, sample 1–3 (**a**–**c**). The collective distribution of the three samples (**d**). The *x*- and *y*-axes are the two discriminant functions.

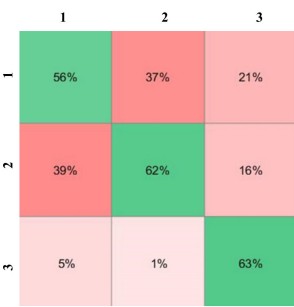

**Figure 6.** Identification results when IMFs were not selected.

As shown in Figure 7, the kurtosis and sensitivity of each IMF in sample 1 showed good consistency, i.e., IMFs with high kurtosis always had higher sensitivity. In addition, Equations (4)–(8) show that the calculation of sensitivity takes into account not only intra-sample correlation but also inter-sample correlation. In this way, the sensitivity for selecting IMFs can be used to increase the differentiation among samples and improve the accuracy of fault identification compared with a single principle such as kurtosis.

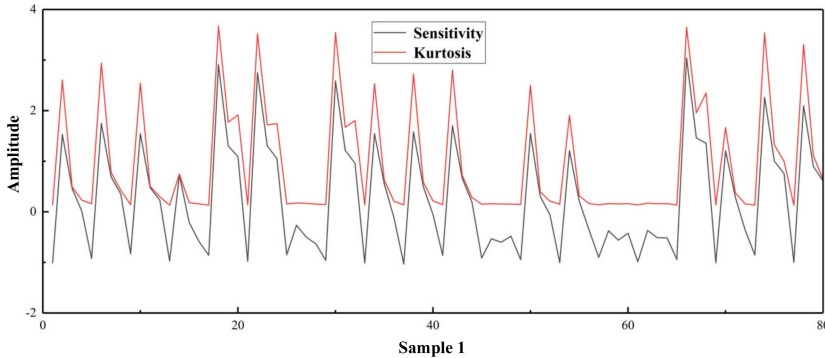

**Figure 7.** Sensitivity and Kurtosis of each IMF of sample 1. The *x*-axis is the number of IMFs and the *y*-axis is the sensitivity value of each IMF.

Considering that the sensitivity of each IMF directly affects the diagnostic results, the Fisher criterion is used to evaluate the diagnostic ability of IMFs, which is helpful to select high-priority IMFs for feature extraction. Sensitivity factor *J* is calculated for each IMF using Equation (8). The values of *J* for 480 IMFs are shown in Figure 8, where the *x*-axis is the number of IMFs and the *y*-axis is the sensitivity value of each IMF. Figure 8 also shows the IMFs obtained from the three-type failure samples.

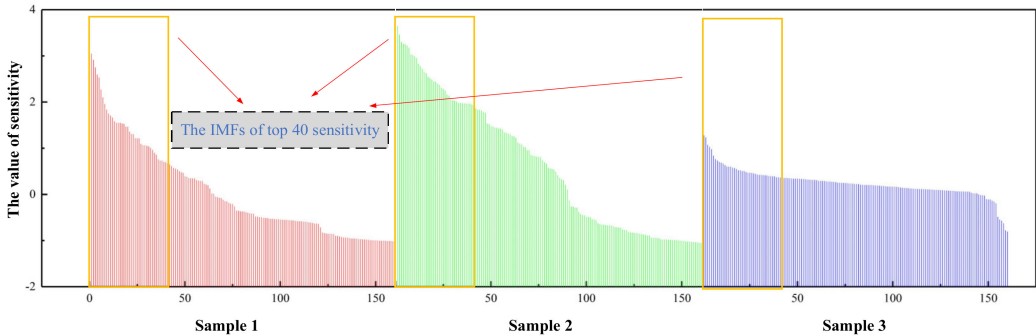

**Figure 8.** Sensitivity ranking of IMFs for the three samples. The *x*-axis is the number of IMFs of each sample and the *y*-axis is the sensitivity value of each IMF.

As shown in Chart 1, the IMFs were obtained after VMD processing for the three fault types. Some of these IMFs had better differentiation and characterization of the different classes of fault information. In addition, not all the IMFs obtained after the VMD pattern decomposition of the samples contained fault information. In contrast, the recognition ability of the extracted features decreased in some VMD modes. Therefore, it will be necessary to find a method to select sensitive and superior IMFs from the original and multimodal VMD signals.

The IMFs obtained from these samples are arranged from smallest to largest sensitivity, as shown in Figure 8. Among these IMFs, redundant IMFs also inevitably led to inaccurate recognition. Therefore, the first 40 IMFs were selected from 480 IMFs to reduce the dimensionality of the input features of the classifier. The sensitivity values of the first 40 IMFs of the three samples are shown in Figure 8.

The IMFs are better classified when they are clearly separated in the feature space. These highly sensitive IMFs can enhance the identification accuracy. After extraction and evaluation, these 40 IMFs are fed into the classifier, thus reducing the computational effort while improving the classification accuracy. The feature distribution of each type of IMF is presented separately in Scheme 2a–c, while Scheme 2d shows the distribution of the three types of IMFs in a uniform feature space.

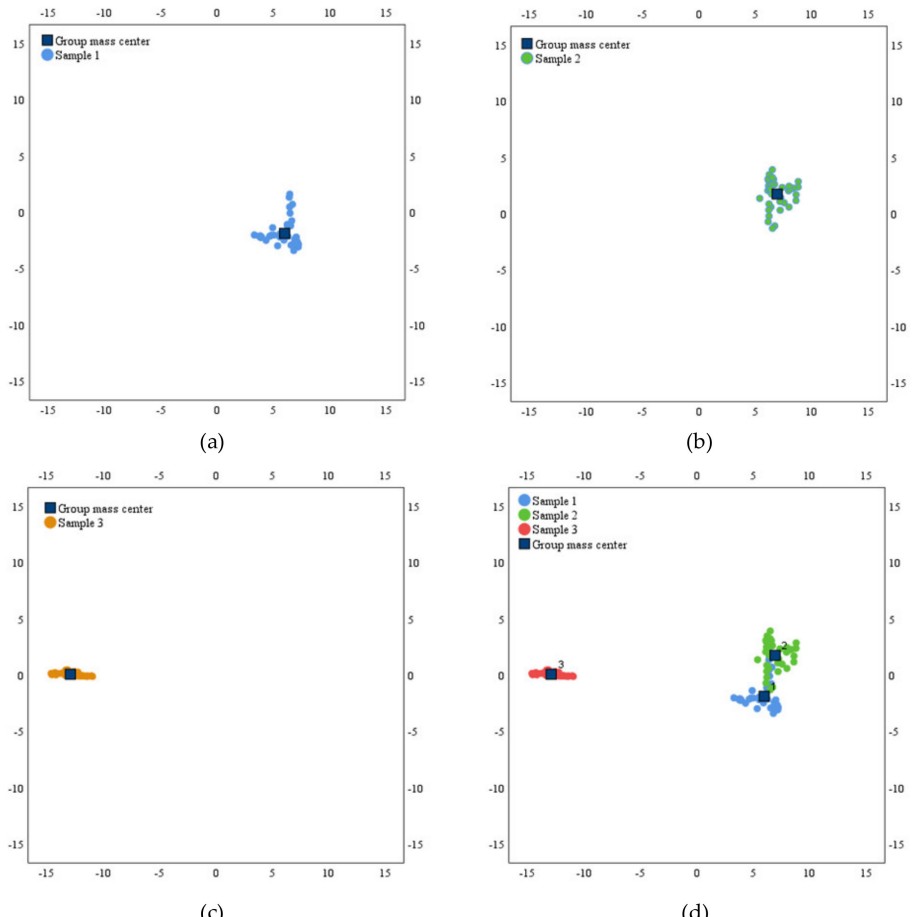

**Scheme 2.** The distribution of selected IMFs in samples 1–3 (**a**–**c**). The collective distribution of the three samples (**d**). The *x* and *y*-axes are the two discriminant functions.

The recognition results for the features of the selected IMFs are shown in Figure 9. It can be seen from the confusion matrix that a high level of fault recognition accuracy was obtained. Specifically, the recognition accuracy for Type 3 reached 100%.

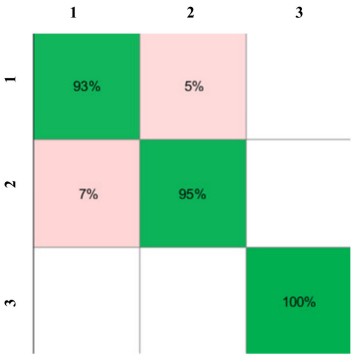

**Figure 9.** Identification results of the proposed method.

*3.2. Comparison with Different Classifiers*

To demonstrate the superiority of SVM in the method proposed in this paper, we used some commonly used classifiers for a comparison, i.e., KNN and DT. The recognition results of different classifiers are shown in the Table 3.

**Table 3.** Number of correctly identified samples.

| Type | Classifiers | | |
|:---:|:---:|:---:|:---:|
| | **SVM** | **TREE** | **KNN** |
| 1 | 37 | 38 | 36 |
| 2 | 38 | 36 | 37 |
| 3 | 40 | 40 | 40 |
| Total | 115 | 114 | 113 |
| Accuracy (%) | 95.8 | 95 | 94.2 |

Overall, SVM had the worst performance compared to the other two classifiers. The performance of DT was slightly better than that of KNN. However, the diagnostic accuracy of Type 2 from DT was too low. One the other hand, the overall accuracy of SVM was 95.8%, which was better than KNN and DT. In addition, SVM's Type 1 and Type 2 recognition accuracy was also better than those of DT and KNN.

*3.3. Discussion*

This paper proposed a contactless and effective fault diagnosis method based on sound signals. The feasibility and superiority of the proposed method were verified by experiments on a folding car mirror. The overall identification accuracy reached 95.8%. The proposed method provides the possibility to realize contactless fault diagnoses based on sound signals, which can provide a form of non-destructive monitoring of mechanical components. At the same time, the experimental environment was influenced by external factors. In order to improve the anti-interference ability of the model, it will be important to explore the robustness and accuracy of the proposed sound signal fault diagnosis method.

**4. Conclusions**

This paper proposed a contactless fault diagnosis method based on VMD and SVM via sound signals. In order to overcome the fault diagnosis accuracy decline caused by IMFs with multiple features sets, the top-ranked IMFs were obtained by FD, and the optimal IMFs for the characterization of fault information were selected through CDF. Then, six dimensionless features were extracted from the selected IMFs which were fed to the classifier as the final features. Experiments on a car mirror folding were conducted to prove the validity of the method proposed. The feasibility and superiority of proposed method were highlighted by comparing different classifiers. Specifically, the three models achieved recognition accuracies of 95.8%, 95%, and 94.2%, respectively. The overall identification accuracy of SVM reached 95.8%, and the SVM obtained the best performance compared to other models.

In future, work will be conducted to establish a model with a higher fault recognition rate.

**Author Contributions:** Methodology, X.Y. and Q.H.; data and experiment, H.Z. and Z.Q.; writing—original draft preparation, Q.H.; writing—review and editing, X.Y. and Q.H.; funding acquisition, B.Z. and X.Y. All authors have read and agreed to the published version of the manuscript.

**Funding:** This work is supported by the National Natural Science Foundation of China under Grants 61803044, Project of the Science and Technology Department of Jilin Province of China under Grand 20200403036SF and 20200301038RQ, and National Natural Science Foundation of China under Grants 61973046.

**Conflicts of Interest:** The authors declare no conflict of interest.

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
