# Peer review of "Sound Based Fault Diagnosis Method Based on Variational Mode Decomposition and Support Vector Machine"

_electronics, doi:10.3390/electronics11152422_

Round 1

Reviewer 1 Report

This article deals with the use of the Variational Mode Decomposition (VMD) and Support Vector Machine (SVM) method to identify defects in mechanical and electrical devices. Such methodology has been intensively researched in recent times, mainly in the context of vibration signals. The authors use these methods for acoustic signals. Although VMD is a fairly simple method of signal decomposition, the authors obtained positive results in identifying the fault signal. This raises the immediate question of the use of more sophisticated blind identification and signal separation techniques such as Independent Component Analisys (ICA) or Nonnegative Matrix Factorization (NMF).

The work contains sufficient results that are worth publishing in a scientific journal. However, there are some aspects of the quality of the work and presentation that authors can improve as indicated in the more detailed comments in the pdf provided.

Reviewer 2 Report

This paper proposed a contactless fault diagnosis method based on Variational Mode Decomposition and Support Vector Machine via sound signals. The experiment on car mirror folding was conducted to prove the validity of the method proposed. The feasibility and superiority of the method proposed are highlighted by comparing different classifiers.

This research has important value. However, before publication, there are still some areas in the article that need to be improved. Below is my comment.

There is no citation for [9] in the text.

Table 1. should be cross-referred in the text.

Figure 3 and its capture are not on the same page.

In the text, figures are referred to as "Figure", "Fig" and "Fig.", "Eq" and "Eq." Use only the one format in the text. Check "Instructions for Authors".

Please clarify Equations 1 and 2 formally!

The equations should be numbered at the end of the rows. Please check this issue!

The number of the figure is missing in row 268.

Some subsections are very short. Please consider expanding them.

Please consider not starting chapters with figures!

Plagiarisms were found in Section 3, Discussion, and Conclusions without citations. Please check this issue!

The section "Discussion" is meaningless in the article because it is a repetition.

"Discussion: Authors should discuss the results and how they can be interpreted from the perspective of previous studies and of the working hypotheses. The findings and their implications should be discussed in the broadest context possible. Future research directions may also be highlighted.

Conclusions: This section is not mandatory but can be added to the manuscript if the discussion is unusually long or complex."

Please check the paper for English editing and typos!

In general, the topic of the article makes a good impression.

Round 2

Reviewer 2 Report

Most of my comments were taken into account and corrections were made. The article looks much better, but minor revision is still required.

Table 1. should be cross-referred in the text (Figure 1. is ok). Please check this issue!

In the text, all figures are referred to as "Figure" except in row 256.

The v2 pdf still has some editing issues:

The equations should be numbered at the end of the rows (Egs. 4, 5, 6, 10, 11)

Please consider not starting chapters with figures (Section 2.5, 3.1)!

Please keep Table 2 in one piece on one page.
